Understanding uremic cardiomyopathy: from pathogenesis to diagnosis and the horizon of therapeutic innovations

Song Qiong 1 2
Wang Pengbo 2
Wu Yunfang 3
Yao Zhuan’e 2
Wang Wei 2
Tang Guangbo ttbocai@126.com 4
Zhang Peng zhangpeng@fmmu.edu.cn 1
1 Department of Nephrology, Shaanxi Provincial People’s Hospital , Xi’an , Shaanxi Province , China
2 Department of Nephrology, The Second People’s Hospital of Shaanxi Province , Xi’an , Shaanxi , China
3 Department of Endocrinology, The First Hospital of Lanzhou City , Lanzhou , Gansu Province , China
4 Key Laboratory of Biomedical Information Engineering of Ministry of Education, School of Life Science and Technology, Xi’an Jiaotong University , Xi’an , Shaanxi Province , China
Capusa Cristina
Electronic publication date: 2025 Mar 31
Publication date: 2025
Volume: 13
Electronic Location ID: e18978
Received 2024 Sep 19; Accepted 2025 Jan 22
Copyright: ©2025 Song et al.
Copyright year: 2025
Copyright holder: Song et al.
License: This is an open access article distributed under the terms of the Creative Commons Attribution License, which permits unrestricted use, distribution, reproduction and adaptation in any medium and for any purpose provided that it is properly attributed. For attribution, the original author(s), title, publication source (PeerJ) and either DOI or URL of the article must be cited.
License URL: https://creativecommons.org/licenses/by/4.0/

Keywords: Uremic cardiomyopathy, Uremic toxins, Inflammation, Oxidative stress, Cardiac hypertrophy

Funding: The authors received no funding for this work.

==============================
Uremic cardiomyopathy (UC) is a significant cardiovascular complication in individuals with end-stage renal disease. This review aims to explore the multifaceted landscape of UC, including the key pathophysiological mechanisms, diagnostic challenges, and current therapeutic approaches. The prevalence of cardiac hypertrophy, as a hallmark of UC, is highlighted and some new insights to its intricate pathogenesis, involving uremic toxins, oxidative stress, and inflammatory responses is elucidated. Diagnostic complexities, including the absence of specific biomarkers, are discussed, and the need for advanced imaging modalities and emerging diagnostic strategies are emphasized. Current therapeutic interventions, although lacking specificity, are addressed, paving its way to the potential future directions in targeted therapies. The review concludes new insights into the critical importance of ongoing research and technological advancements which will enhance early detection, precision treatment, and ultimately improve outcomes for individuals with UC.

Introduction

Uremic cardiomyopathy (UC) is a distinctive form of cardiomyopathy, deriving from chronic kidney dysfunction, and it is the leading cause of mortality in progressive chronic kidney disease (CKD) (Husain-Syed et al., 2015; Sárközy et al., 2018). Cardiovascular-related deaths account for approximately 40% to 50% of all fatalities in patients with severe CKD 4 and those with end-stage renal disease (ESRD), whereas they represent about 26% in individuals with normal kidney function (Thompson et al., 2015; Webster et al., 2017). UC is primarily characterized by diastolic dysfunction, with left ventricular hypertrophy (LVH) as a key feature, often accompanied by increased ventricular thickness, arterial stiffening, coronary atherosclerosis, and coronary artery calcification (Wang & Shapiro, 2019).

The actual prevalence of UC is influenced by various factors, including geographical distribution, complexity of diagnostic criteria, and diagnostic capabilities. Despite limited epidemiological data, UC’s significance is underscored by its links to cardiac fibrosis, heart failure, insulin resistance, and a higher incidence of sudden cardiac death, making it a leading cause of mortality in progressive CKD, particularly in ESRD patients (Hu et al., 2017). The intricate pathogenesis involves classical and renal-specific factors, including pressure and volume overload and the uremic state itself. Left ventricular volume overload is influenced by factors such as arteriovenous fistulas, anemia, and hypervolemia (Law et al., 2023). Aoki et al. (2005) conducted endomyocardial biopsies on 40 cases of ESRD patients exhibiting reduced left ventricular ejection fraction but no evidence of coronary artery disease, revealing significant cardiac fibrosis and indicating that LVH and congestive heart failure are more significant contributors to mortality in this group than coronary artery disease. By 2030, an estimated 2 million people in the USA will require dialysis for ESRD, with around 70% of these patients exhibiting LVH. Key risk factors for UC include uremic toxins, anemia, hypervolemia, oxidative stress, inflammation, insulin resistance, and chronic kidney disease-mineral and bone disorder (CKD-MBD).

While standard medications are available for controlling these risk factors, cardiovascular complications, particularly UC, continue to worsen, highlighting several critical issues in both research and clinical practice (Wang & Shapiro, 2019). These issues include: (1) Unclear mechanisms: the precise molecular mechanisms driving UC are not fully understood, with much of the current research focusing on isolated pathophysiological pathways. There is a notable lack of comprehensive, interdisciplinary studies that integrate genomics, transcriptomics, epigenomics, metabolomics, and other omics technologies to uncover the complex molecular networks at play in UC. (2) Lack of early diagnostic biomarkers: there is no reliable biomarker for the early detection of UC, which leads to delayed diagnosis and progression to advanced stages of myocardial injury before intervention can occur. This highlights a critical gap in our ability to monitor disease onset and progression in CKD patients. (3) Limited effectiveness of current medications: although there are treatments available for managing the underlying conditions associated with UC, no single pharmacological agent has proven to be highly effective in significantly improving patient outcomes. Current medications often fail to address the complex and multifactorial nature of UC, underscoring the need for more targeted and effective therapies. (4) Limited intervention strategies: at present, there are no well-established clinical interventions that can effectively reverse or prevent the myocardial damage caused by UC. The lack of effective therapeutic options further complicates the management of cardiovascular complications in CKD patients. This review aims to systematically present the latest research on the multifactorial mechanisms involved in UC, focusing on the role of uremic toxins, chronic inflammation, oxidative stress, calcium-phosphate metabolism disturbances. We will also discuss emerging strategies and methods, including multi-omics technologies, which combined with AI-based approaches, can help to uncover the complex pathophysiology of UC and provide new insights for the discovery of early biomarkers, diagnosis, and the design of molecular-targeted therapies. By exploring these innovative approaches, this review seeks to propose novel directions for future research aimed at improving early diagnosis and developing more effective, personalized interventions for UC.

Survey Methodology

To ensure comprehensive and unbiased coverage of the literature on UC, a systematic search was conducted using multiple authoritative databases. The primary databases and search engines used included PubMed, Google Scholar, and Web of Science. The search terms and combinations included, but were not limited to: “uremic cardiomyopathy,” “cardiac hypertrophy,” “uremic toxins,” “oxidative stress,” “inflammation,” “diagnostic challenges,” and “therapeutic approaches.” Additionally, medical subject headings (MeSH) terms were employed to enhance the breadth and accuracy of the search.

Inclusion criteria for the literature were: (1) original research articles, systematic reviews, and relevant publications from the last 20 years; (2) studies addressing the pathophysiology, diagnostic challenges, or therapeutic strategies of UC; and (3) articles published in peer-reviewed journals. Exclusion criteria included: (1) non-English language publications; (2) articles not peer-reviewed; and (3) studies not directly related to UC.

Pathogenesis of uremic cardiomyopathy

Previous studies have identified risk factors for UC, such as uremic toxins, secondary hyperparathyroidism, abnormal bone metabolism, and insulin resistance. However, recent research shows that UC is also characterized by various cardiac metabolic disorders and altered microRNAs (miRNAs). Figure 1 summarizes the pathophysiological pathways from kidney disease to heart failure.

Figure 1 Pathophysiological pathway from renal disease to heart failure.

This diagram depicts the complex pathways leading from end-stage renal disease to uremic cardiomyopathy. Multiple factors, including uremic toxins, metabolic disturbances, and ionic imbalances, contribute to cardiac damage and dysfunction. The interplay of these mechanisms ultimately results in reduced cardiac efficiency and structural changes characteristic of uremic cardiomyopathy.

Uremic toxins accumulation

The accumulation of uremic toxins results from progressive renal function decline, leading to the inefficient elimination of toxins and waste substances from the body. Examples of these toxins include urea, creatinine, phenols, and indoxyl sulfate (IS).

Elevated uremic toxin levels significantly contribute to endothelial dysfunction, which plays a role in cardiovascular diseases such as atherosclerosis and thrombotic events. The accumulation of these toxins can trigger apoptosis in cardiac muscle cells, leading to metabolic abnormalities, oxidative stress, and inflammation, ultimately resulting in myocardial damage (Wojtaszek et al., 2021).

Substances like phosphate and urea are closely linked to cardiovascular events. Phosphate has a multifaceted impact, directly influencing vascular smooth muscle cells (VSMCs) through complex calcium/phosphate interactions, and indirectly through the fibroblast growth factor 23 (FGF23)/Klotho axis and hyperparathyroidism (PTH) pathways (Disthabanchong, 2018).

These toxins also contribute to other cardiac conditions such as fibrosis, arrhythmias, and hypertrophy. IS is linked to atrial fibrillation, intensifying fibrosis and hypertrophy through increased oxidative stress (Pieniazek et al., 2022). Elevated levels of phosphate and urea are also associated with cardiac hypertrophy and fibrosis. High urea levels contribute to systemic inflammation, thickening subepicardial arteries, and collectively exacerbating diastolic dysfunction (Carmona et al., 2011).

Abnormal metabolism of cardiomyocytes

Metabolic dysregulation within cardiac muscle cells plays a crucial role in the pathogenesis of UC, involving impaired mitochondrial function, altered myocardial substrate utilization, and changes in the function and expression of metabolic transport proteins (Nguyen & Schulze, 2023). The accumulation of uremic toxins has been found to hinder mitochondrial function in cardiac muscle cells, leading to oxidative stress and inflammatory responses, both contributing significantly to the advancement of myocardial damage (Sun et al., 2017). This metabolic disruption, involving glucose, lipid, and amino acid metabolism, contributes to myocardial damage progression, leading to cellular injury and apoptosis.

In a study, endothelial dysfunction and recurrent reductions in myocardial blood flow during hemodialysis may contribute to the development of metabolic abnormalities, potentially with regional variations (Kawano, Kudo & Maemura, 2021).

Furthermore, metabolic irregularities in cardiac muscle cells may impact cardiac neuroendocrine function, potentially leading to a spectrum of disorders, including arrhythmias, myocardial hypertrophy, and impaired cardiac diastolic function (Nguyen & Schulze, 2023).

Inflammation and oxidative stress

Oxidative stress (OS), defined as the disequilibrium between excessive prooxidant activities and the defense mechanisms of antioxidants, induces metabolic dysregulation, leading to the oxidation of lipids, proteins, and nucleic acids. Reactive oxygen species (ROS) and reactive nitrogen species (RNS) are responsible for oxidative damage to cells, tissues, and organs (Di Meo et al., 2016).

In the early phases of renal disease, OS is evidently heightened, characterized by an imbalance in ROS production and degradation, including uncoupling of nitric oxide production. This imbalance contributes to endothelial damage, serving as a risk factor for congestive heart failure (CHF) by affecting cardiac microcirculation and inducing endothelial dysfunction (Sárközy et al., 2018).

CKD creates a persistent low-grade inflammatory state, involving cytokines such as C-reactive protein (CRP), interleukin-6 (IL-6), pentraxin-3, tumor necrosis factor-alpha (TNF-alpha), adhesion molecules, and fibrinogen (Filiopoulos & Vlassopoulos, 2009).

Uremic toxins, including IS, p-cresol (PC), and p-cresol sulfate (PCS), are implicated in cardiovascular diseases among CKD patients, primarily inducing endothelial dysfunction and vascular calcification through OS and inflammation (Ribeiro et al., 2023). Uremic toxins are also implicated in neurodegenerative disorders associated with CKD, escalating oxidative stress in glial cells and downregulating the transcription factor Nrf2, leading to decreased expression of cytoprotective and antioxidant enzymes. Meanwhile, dialysis contributes to increased OS through various mechanisms, including the use of bioincompatible membranes and fluids, dialysate contamination with bacterial endotoxins, and occult infections (Rodríguez-Ribera et al., 2017).

Insulin resistance and metabolic abnormality

Insulin resistance, a common occurrence in patients with CKD, stands as an independent risk factor for cardiac disease in this population (Sebastian, Padda & Johal, 2024). Factors contributing to this disruption include increased angiotensin II, inflammation, metabolic acidosis, and uremic toxins (Thomas, Zhang & Mitch, 2015).

Underlying insulin resistance in patients with uremia disrupts the balance of insulin’s pleiotropic effects. Metabolic acidosis in CKD contributes to uremic insulin resistance, as correcting pH has been shown to improve the insulin sensitivity of glucose metabolism (Spoto, Pisano & Zoccali, 2016).

Insulin resistance affects the heart through various mechanisms, by downregulating sodium-calcium exchange, decreasing myosin ATPase activity, and upregulating angiotensin II, ultimately resulting in reduced cardiac efficiency (Singh et al., 2008).

In uremia, insulin resistance may exacerbate metabolic remodeling in the heart by limiting the uptake and metabolism of glucose, predisposing the uremic heart to progress into chronic energy deficiency and eventually heart failure.

Cardiac structure abnormality

UC is characterized by a combination of LVH, diastolic dysfunction, capillary rarefaction, endothelial dysfunction, and cardiac fibrosis. These factors collectively contribute to the development of heart failure (HF) (Sárközy et al., 2018). These findings highlight the significance of the underlying fibrotic process and myocardial edema as fundamental mechanisms in the pathophysiology of UC. Initially viewed as a beneficial adaptive response in the early phases of UC, LVH becomes maladaptive as progressive LV overload induces detrimental changes in cardiomyocytes, ultimately leading to cell death (Garikapati et al., 2021).

UC-associated LV hypertrophy results from complex pressure overload, volume overload, and the uremic state itself. Some authors suggest a higher proportion of deaths in these patients attributed to LVH and congestive heart failure, surpassing those caused by coronary artery disease (Alhaj et al., 2013). Within the uremic heart, elevated expression of profibrosis mediators such as transforming growth factor beta (TGFβ) and connective tissue growth factor (CTGF) leads to increased collagen levels, contributing to interstitial fibrosis. This, in turn, contributes to diastolic dysfunction, ventricular stiffness, and cardiac dysrhythmias (Lekawanvijit et al., 2012).

CKD-MBD

Chronic kidney disease-mineral and bone disorder (CKD-MBD) is a significant contributor to UC. A comprehensive understanding of CKD-MBD’s role in cardiomyopathy development necessitates the analysis of key factors such as PTH, phosphate, fibroblast growth factor-23 (FGF-23), and Klotho. The crucial role of phosphate in CKD-MBD is notable, and its toxicity is implicated in contributing to cardiovascular mortality (Ritter & Slatopolsky, 2016).

CKD patients often experience deficiencies in renal excretory capacity, leading to significant upsurges in FGF-23 levels as a compensatory response to preserve phosphate levels. These elevated FGF-23 levels have been associated with cardiovascular disease (CVD) and mortality in a dose-dependent fashion (Vergaro et al., 2018). The kidney-expressed transmembrane protein, Klotho, acts as a co-receptor for FGF-23, playing a role in mediating phosphate regulation. Emerging data suggests that FGF-23 and Klotho collaboratively contribute to UC in a coadjutant manner (Memmos & Papagianni, 2021).

The mechanism underlying myocardial dysfunction in hyperparathyroidism is believed to involve the uncoupling of oxidative phosphorylation, resulting in reduced cellular ATP concentrations and impaired calcium extrusion. This, in turn, leads to calcium overload in cardiomyocytes (Ahmadmehrabi & Tang, 2018).

MicroRNAs and molecular regulation

Specific microRNAs (miRNAs) have been identified as significant roles in the development and progression of UC (Table 1). UC is characterized by myocardial remodeling, fibrosis, and impaired response to injury. Among the miRNAs, miR-26a is decreased in CKD mouse hearts and helps alleviate cardiac fibrosis and improve cardiac function when administered (Wang et al., 2019).

Table 1 Roles of specific microRNAs in uremic cardiomyopathy (UC).

miRNA	Expression changes in UC	Mechanisms and effects	References	
miR-26a	Decreased	Alleviates cardiac fibrosis and improves cardiac function by enhancing insulin sensitivity and reducing profibrotic factors when administered	Wang et al. (2019)	
miR-30	Decreased	Modulates autophagy, apoptosis, and oxidative stress in cardiomyocytes, associated with renal damage and CKD-induced cardiac hypertrophy	Bao et al. (2021)	
miR-29b-3p	Decreased	Exerts antifibrotic effect by acting on the Na/K-ATPase signaling pathway, downregulation leads to increased cardiac fibrosis	Drummond et al. (2016)	
miR-212/132	Elevated	Promotes left ventricular hypertrophy (LVH) and heart failure by repressing FoxO3 under pressure overload, potentially follows different pathways in CKD	Ucar et al. (2012)	
miR-208	Not specified	Modulates α-myosin heavy chain (α-MHC) and myocardial fibrosis, crucial for heart remodeling, regulated by thyroid hormones	Prado-Uribe et al. (2013)	

The miR-30 family plays a role in modulating autophagy, apoptosis, and oxidative stress in cardiomyocytes, with lower expression linked to renal damage and CKD-induced cardiac hypertrophy (Bao et al., 2021). miR-29b-3p has an antifibrotic effect via the Na/K-ATPase signaling pathway, where its downregulation leads to increased cardiac fibrosis (Drummond et al., 2016). miR-212/132 promotes LVH and heart failure by repressing FoxO3 under pressure overload conditions, with increased levels observed in CKD (Ucar et al., 2012).

Other notable miRNAs include miR-208, which modulates α-myosin heavy chain (α-MHC) and myocardial fibrosis, crucial for heart remodeling and regulated by thyroid hormones (Prado-Uribe et al., 2013). These miRNAs are promising biomarkers for diagnosis, prognosis, and therapeutic targets in UC, with ongoing research aimed at understanding their complex molecular mechanisms to improve care for CKD and ESRD patients.

Diagnostic tools and cardiac manifestations in uremic cardiomyopathy

In current clinical practice, numerous uncertainties affect the precise diagnosis of cardiomyopathy, leading to a high rate of misdiagnosis. These challenges include phenotypic similarities, diverse clinical symptoms, complex genetic backgrounds, limitations in diagnostic technology, insufficient expertise, lack of screening and testing, and limited treatment options. Therefore, making full use of various non-invasive diagnostic techniques such as echocardiography, CT scans, and cardiac magnetic resonance imaging (CMR) is crucial for accurately diagnosing and guiding the treatment of cardiomyopathy by comprehensively assessing its morphology, function, and histological changes (McDonagh et al., 2024). The electrocardiograph (ECG) stands as the most widely available, least invasive, and least expensive method for evaluating left ventricular (LV) dysfunction, as well as identifying volume and pressure abnormalities in patients with CKD. Typical ECG findings include the presence of Q waves, dynamic ST segment changes, prolonged QRS duration, tachycardia, and left and right atrial enlargement (Shafi et al., 2017). On the other hand, multiple imaging modalities, including transthoracic echocardiography (TTE), cardiac CT, and cardiac magnetic resonance imaging (CMRI), can be employed for cardiac assessments in patients with CKD. The extent of fibrosis serves as a robust biomarker for cardiovascular death (McIntyre, John & Jefferies, 2008).

TTE stands as a readily available, cost-effective, and noninvasive modality that allows detailed observation of cardiac structures, particularly LVH. TTE is a well-established method for assessing left ventricular mass, commonly utilized to provide prognostic information or as an endpoint in studies related to CVD risk (Badve et al., 2016). Cardiac computed tomography (CT) scanning is applicable in UC, providing the capability to delineate regions of myocardial fibrosis. Cardiac MRI (cMRI) is a valuable tool that offers information on various cardiac structural and functional parameters, encompassing coronary arterial flow, perfusion, myocardial scarring, and interstitial fibrosis. Consequently, due to its available and practical, cMRI has become the standard for cardiac imaging. Cardiac magnetic resonance imaging results are less influenced by left ventricular volume changes and have been proven to be superior to echocardiography in the diagnosis of patients with ESRD (Sobh et al., 2021). CMR T1 and T2 mapping techniques expanded our ability to provide early identification and sensible tracking of disease progression in UC.

There are some biochemical markers in the blood plasma that can reflect cardiac function, such as N-terminal pro-brain natriuretic peptide (NT-proBNP), high-sensitivity cardiac troponin T (hs-troponin T), galectin-3 (Galactin-3), etc. These plasma biomarkers can play an auxiliary role in assessing cardiovascular risk in patients with ESRD, but they still cannot replace imaging examinations (Ponikowska et al., 2022). Figure 2 illustrates the primary diagnostic tools used in the assessment of UC.

Figure 2 Main diagnostic tools for uremic cardiomyopathy.

This comprehensive approach combines traditional diagnostic methods with advanced imaging techniques to provide a thorough assessment of cardiac structure and function in patients with uremic cardiomyopathy.

Due to the diagnostic complexity of UC, the current biomarkers still lack sufficient specificity and sensitivity, and there is a lack of specific markers, which limits early diagnosis and treatment of the disease. Based on existing AI technologies, we aim to explore how multi-omics data and artificial intelligence can provide potential solutions to address this issue.

Potential applications and challenges of AI in the diagnosis and biomarker discovery of uremic cardiomyopathy

AI-based image recognition is transforming the diagnosis of UC, enabling faster and more accurate analysis of cardiac imaging data than traditional methods. Deep learning models, trained on extensive datasets of UC cases, can minimize human error and accelerate diagnosis, especially in clinical settings where early detection is vital for treatment planning (Elias et al., 2024; Glass et al., 2022). This section explores the potential group designs, data categories for AI analysis for identifying early biomarkers, and challenges that may arise during implementation.

Control group design for early biomarkers

To identify early biomarkers, appropriate control group design is crucial. The design should consider the disease’s early characteristics and potential confounders. Some options include:

• Healthy control vs. patient groups: Age-, sex-, and clinical feature-matched healthy volunteers serve as a baseline, while patient groups can be divided into early-stage CKD patients (without significant cardiac symptoms) and end-stage patients (with obvious uremic cardiomyopathy symptoms).

• CKD control group: CKD patients without cardiac abnormalities can be used to distinguish kidney-specific effects from cardiomyopathy-related changes. A normal kidney function group can further isolate kidney and heart health factors.

• Disease progression groups: Patients with UC can be grouped by acute and chronic phases to identify biomarkers relevant to early vs. advanced stages.

• Clinical and laboratory-based groups: Clinical risk stratification groups can focus on high-risk patients based on factors like blood pressure or proteinuria. Biochemical marker-based groups can be stratified by kidney and cardiac markers to identify early cardiac damage.

• Dynamic monitoring vs. cross-sectional groups: A dynamic monitoring group can track longitudinal changes in biomarkers, while a cross-sectional group provides a snapshot for quick identification of early biomarkers.

Data for AI-based disease recognition in uremic cardiomyopathy

AI-based disease recognition can benefit from various data types:

1. Blood biomarkers: Uremic toxins, inflammatory markers, and cardiac biomarkers help assess disease severity and progression.

2. Fluid biomarkers: Data from body fluids (e.g., urine, peritoneal fluid) can provide insights into systemic effects of uremic toxins.

3. Cardiac imaging: MRI, CT, and PET scans offer comprehensive data on heart structure and function.

4. Genetic testing: AI can use genetic data to identify at-risk patients before symptoms arise.

5. Myocardial biopsy: Histological data from myocardial biopsies can enhance AI models by revealing cellular changes in the heart muscle.

Challenges and solutions

Several challenges need to be addressed to optimize AI in diagnosing UC:

1. Data quality and variability: AI models require high-quality, consistent data. Variability in imaging, demographics, and disease stages can limit generalizability. Larger, more diverse datasets and standardized protocols are needed to improve consistency.

2. Interpretability and clinical integration: Deep learning models are often “black-box,” making predictions difficult to interpret. Developing explainable AI models will enhance clinician trust and usage.

3. Regulatory and ethical issues: Data privacy, validation, and regulatory concerns must be addressed to ensure AI systems are safe, reliable, and compliant with clinical standards.

Therapeutic approaches and emerging insights in uremic cardiomyopathy

Angiotensin-converting-enzyme inhibitors and angiotensin II receptor blockers

At present, angiotensin-converting-enzyme inhibitors (ACEi) and angiotensin II receptor blockers (ARBs) are the primary therapeutic drugs for UC, serving to counteract the effects of the renin-angiotensin-aldosterone system (RAAS). ACEi/ARBs contribute to improved vascular endothelial function, enhanced fibrinolysis, and the antagonism of vascular smooth-muscle cell proliferation and plaque rupture (Poskurica & Petrović, 2014).

Sacubitril/valsartan functions as an angiotensin-receptor neprilysin inhibitor. By acting as an antagonist to angiotensin II receptors, sacubitril/valsartan can impede the activation of the RAAS, thereby reducing vasoconstriction and cardiomyocyte fibrosis. Additionally, neprilysin inhibition prevents the breakdown of natriuretic peptides, thus enhancing their activity. This leads to vasodilation, increased natriuresis, and decreased cardiac workload (Vaduganathan et al., 2020). A post hoc analysis of the PARAGON-HF trial revealed that serum levels of NT-proBNP and cardiac troponin T were notably reduced in the sacubitril/valsartan group compared to the valsartan group. This suggests that sacubitril/valsartan is more effective in reducing ventricular hemodynamic stress and safeguarding against cardiomyocyte damage (Chandra et al., 2019). Furthermore, sacubitril/valsartan significantly elevates circulating cGMP levels, suppresses the expression of gene programs associated with cardiac fibroblasts, diminishes fibroblast activation and proliferation, enhances myocardial flexibility, and slows down cardiac hypertrophy and remodeling (Mochel et al., 2019).

Vericiguat

Medications targeting the nitric oxide (NO), soluble guanylate cyclase (sGC), and cyclic guanosine monophosphate (cGMP) pathway for treating heart failure are gaining increasing attention. As a secondary messenger, cGMP functions include vasodilation, inhibition of platelet aggregation, suppression of smooth muscle proliferation, and anti-cardiac remodeling, anti-proliferative, and anti-inflammatory effects (Murad, 2006). Vericiguat, an sGC stimulator, serves as a novel targeted therapy for heart failure. It operates by activating the NO-sGC-cGMP pathway, resulting in vasodilation and amelioration of cardiac remodeling (Armstrong et al., 2020). However, additional research is needed to explore the feasibility of adding vericiguat to standard heart failure treatment in patients with an eGFR < 30 mL min−1 (1.73 m2)−1.

Vitamin D

Furthermore, the significance of vitamin D is becoming increasingly recognized. Supplementing vitamin D in hemodialysis patients led to a notable decrease in PTH levels, an elevation in LV fiber fraction shortening, and an increase in the mean velocity of LV fiber shortening (D’Amore et al., 2017).

Insulin sensitizers

Recent data suggests that insulin sensitizers, such as PPAR-gamma agonists like rosiglitazone and pioglitazone, may partially restore lost insulin-mediated protection in the uremic population. These drugs offer multifaceted benefits, including anti-inflammatory, antihypertensive, anti-proteinuric, and insulin-sensitizing effects (Hung & Ikizler, 2011).

Renal replacement therapy

Regardless of pharmacotherapy, the ultimate treatment for UC typically involves renal replacement therapy, including hemodialysis, peritoneal dialysis, or renal transplantation. Conventional hemodialysis is the most common treatment for UC. Numerous studies have shown that hemodialysis enhances reverse cardiac remodeling and reverses certain clinical consequences of UC, making it a crucial treatment (Adhyapak & Iyengar, 2011). Indeed, the strikingly elevated incidence of cardiovascular events and the severity of LVH have shown a significant reduction following successful kidney transplantations (Diakos et al., 2016).

Other candidate drug targets

Neuregulin-1β (NRG-1β) is a stress-induced paracrine transmembrane growth factor released by endothelial cells. It plays a crucial role in embryonic cardiac development. Dysregulation of NRG-1β expression has also been associated with various cardiovascular diseases, including myocardial infarction and HF (Geissler, Ryzhov & Sawyer, 2020). The administration of recombinant human NRG-1β (rhNRG-1β) protein has been shown to protect against myocardial ischemia/reperfusion injury and cardiac fibrosis, as well as to promote cardiac repair following myocardial infarction (Liu et al., 2006). Sárközy et al. (2023) conducted the first study demonstrating that treatment with rhNRG-1β can prevent the progression of renal dysfunction and UC, and also reduce circulating levels of uremic toxins in a rat model of chronic kidney disease (CKD). Therefore, rhNRG-1β may represent a novel and promising therapeutic strategy in translational studies aimed at treating UC.

Clinicaltrials

Through consulting relevant databases (https://clinicaltrials.gov), we have identified several clinical trials on pharmacological treatment for UC. One completed single-center, randomized controlled trial investigated the impact of intravenous L-carnitine (1,000 mg per dialysis session) on myocardial fatty acid metabolism in hemodialysis patients over the course of one year before and after administration. Additionally, a recruitment is currently underway to explore the efficacy and safety of colchicine as an anti-inflammatory treatment for UC.

Shortcoming

Currently, the efficacy of available drug treatments for UC is limited, and there are few clinical trials for these drugs. We aim to explore potential drug targets and drug discovery strategies to address this issue.

Strategies for developing therapeutic drugs for UC

Developing effective therapies for UC is challenging due to the disease’s complexity and multifactorial nature. UC is driven by uremic toxin accumulation, chronic inflammation, oxidative stress, calcium-phosphate disturbances, and immune dysfunction. Targeting these mechanisms offers potential, but the lack of specific drugs highlights the need for new strategies. Below, we discuss approaches for UC drug development, focusing on neutralizing uremic toxins, identifying targets, and exploring new drug strategies:

(1) Neutralizing uremic toxins: Uremic toxins contribute to UC by inducing oxidative stress, inflammation, and endothelial dysfunction. Neutralizing or removing these toxins is a promising approach.

• Targeting specific toxins: Certain uremic toxins, like indoxyl sulfate and AGEs, accelerate UC. Developing inhibitors or binders to neutralize these toxins could mitigate their effects. Adsorbents, such as activated charcoal or resins, may be used during dialysis to remove these toxins from the bloodstream.

• Enzyme-based detoxification: Enzyme therapies and gut microbiota modulation via probiotics could degrade toxins like indoles and phenols, reducing their absorption and accumulation.

• Dialysis enhancements: Improving dialysis techniques, like high-flux hemodialysis, could enhance toxin removal, especially for larger molecules or toxins that affect the heart.

(2) Target identification for UC: Identifying molecular targets involved in UC is essential for developing specific therapies. Key targets include:

• Inflammatory pathways: Inflammation plays a critical role in UC. Targeting inflammatory cytokines (e.g TNF-α, IL-6) or their receptors could reduce myocardial injury (Yamaguchi et al., 2022). Anti-inflammatory agents, such as monoclonal antibodies, are being explored.

• Oxidative stress regulators: Targeting the Nrf2 pathway or sirtuins may reduce oxidative stress and improve heart function, offering potential for UC treatment (Liu et al., 2022).

• Calcium-phosphate disturbances: Targeting the FGF23-Klotho pathway or phosphate transporters could restore calcium-phosphate balance and prevent vascular calcification, improving cardiac function (Grabner & Faul, 2016).

• Fibrosis and remodeling: Targeting fibroblast activation and collagen deposition may prevent myocardial stiffening. Drugs inhibiting TGF-β or the SMAD pathway could reduce fibrosis and enhance myocardial compliance.

(3) Repurposing existing drugs: Repurposing FDA-approved drugs for other conditions offers a promising strategy for UC treatment. Drugs targeting inflammation, oxidative stress, or fibrosis—like statins, ACEi and ARBs—may be effective in UC. Machine learning and big data analysis of existing drug libraries could accelerate the repurposing process for UC.

Enhancing practical value for clinicians in managing chronic kidney disease and uremic cardiomyopathy patients

Managing CKD and UC is challenging due to the interplay of uremic toxins, cardiovascular complications, and comorbidities. A clearer framework for patient management can improve outcomes through informed treatment decisions, early intervention, and enhanced monitoring.

Early diagnosis, prognosis, and monitoring

Early detection of UC in CKD patients is difficult due to a lack of specific biomarkers and guidelines. Integrating multi-omics data, machine learning, and imaging can aid early diagnosis, allowing timely interventions to slow UC progression. Nephrologists monitor renal function, cardiologists assess cardiac involvement, and diagnostic specialists use advanced imaging to detect abnormalities. Real-time biomarkers and monitoring technologies will enable proactive care and reduce complications.

Targeted therapeutic strategies and interdisciplinary collaboration

Developing novel drugs and repurposing existing ones will offer specific treatment options targeting inflammation, oxidative stress, fibrosis, and toxin neutralization. Clear treatment algorithms will assist nephrologists and cardiologists in managing UC alongside CKD. Pharmacologists assess drug efficacy and safety, while a multidisciplinary approach involving nephrologists, cardiologists, endocrinologists, and pharmacologists will improve patient care by addressing renal, cardiac, and metabolic aspects.

Personalized medicine and optimized treatment plans

Personalized medicine allows treatment plans based on genetic, molecular, and clinical data. Endocrinologists manage metabolic disturbances, while nephrologists and cardiologists focus on kidney and heart care. Personalized approaches help optimize interventions like dialysis and adjust dietary regimens. These strategies improve drug efficacy and reduce adverse effects.

Education, training, and continued research

Clinicians must stay updated on the latest research, therapies, and technologies related to UC and CKD. Incorporating innovations into clinical practice guidelines and continuous training programs ensures that healthcare providers are equipped with the necessary tools to manage these conditions effectively.

Conclusions

The goal of studying UC, characterized by left ventricular hypertrophy, is to find new therapeutic targets. Currently, there is no specific treatment for UC, and existing treatments may be insufficient. Addressing this gap is a crucial concern for future diagnosis and treatment.

We have reviewed the pathogenesis, diagnosis, and pharmacological treatment of UC. Addressing the current challenges, including the lack of early biomarkers, the shortage of effective therapeutic drugs, and the absence of standardized clinical management pathways, we propose solutions based on multi-patient group controls and multi-dimensional clinical data, combined with machine learning and artificial intelligence technologies, to identify specific biomarkers and provide key implementation strategies. Additionally, we discuss potential drug targets and drug development strategies. Finally, we offer comprehensive, multidisciplinary recommendations for the clinical management of UC. We hope that our review will contribute to advancing this field. However, research and treatment of UC still face significant challenges, including the complexity of mechanistic studies, and the large data requirements, long timelines, and high costs associated with biomarker discovery and drug development.

Additional Information and Declarations

Competing Interests

Author Contributions

Data Availability

The authors declare there are no competing interests.

Qiong Song performed the experiments, authored or reviewed drafts of the article, and approved the final draft.

Pengbo Wang performed the experiments, authored or reviewed drafts of the article, and approved the final draft.

Yunfang Wu analyzed the data, prepared figures and/or tables, and approved the final draft.

Zhuan’e Yao analyzed the data, prepared figures and/or tables, and approved the final draft.

Wei Wang analyzed the data, prepared figures and/or tables, and approved the final draft.

Guangbo Tang conceived and designed the experiments, authored or reviewed drafts of the article, and approved the final draft.

Peng Zhang conceived and designed the experiments, authored or reviewed drafts of the article, and approved the final draft.

The following information was supplied regarding data availability:

This is a literature review.

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
