# Peer review of "Understanding uremic cardiomyopathy: from pathogenesis to diagnosis and the horizon of therapeutic innovations"

_PeerJ, doi:10.7717/peerj.18978_

## Round 0.1 · original submission · Major Revisions

A better structure of the manuscript is needed to increase its informative and teaching role. Th authors should use sub-headings to separate topics.
In addition, more interpretation of the basic data is required, with the addition of newer references and comments about the clinical applicability and potential therapeutic targets.

·

Basic reporting

1)Yes, the manuscript is written in clear and unambiguous professional English throughout,
2)To ensure the manuscript aligns with the claim of referencing recent literature (within the last 10 years), it's advisable to review and update the references, particularly those related to clinical epidemiology and other rapidly evolving fields. While some foundational studies are often necessary and valuable, replacing outdated references (like those from 1999) with more recent studies will strengthen the manuscript's relevance and credibility. This will better support the assertion that the article draws from the latest available research.
3) The review appears to be of broad and cross-disciplinary interest, particularly relevant to fields including cardiology, nephrology, and translational medicine, as it addresses the significant cardiovascular implications of end-stage renal disease, specifically uremic cardiomyopathy (UC). By covering pathophysiological mechanisms, diagnostic challenges, and therapeutic approaches, it provides a comprehensive overview that can appeal to a wide range of researchers and clinicians.
4) Yes, the field of uremic cardiomyopathy (UC) has been reviewed in recent years, especially as research on end-stage renal disease and cardiovascular complications has expanded. But this article failed to address
Diagnostic complexities such as the lack of specific biomarkers and the role of advanced imaging, which is crucial for both early detection and tailored treatment strategies.
Emerging targeted therapies and the future direction of UC treatment, which are timely topics as precision medicine gains traction.
Cross-disciplinary insights that make it accessible to a wider audience, including specialists in cardiology, nephrology, and clinical diagnostics.
5)The introduction does not fully achieve its purpose of effectively engaging the audience or clearly stating the motivation for the review. Here are the primary issues:

Lacks Clear Context and Motivation: Although it provides detailed information on UC, it does not offer a compelling rationale for why a review is necessary. The introduction would benefit from explicitly addressing recent advancements in UC research, the current knowledge gaps, and how this review intends to fill them.
Overemphasis on Basic Information: The text focuses extensively on introductory facts about UC but lacks an overarching narrative that connects these facts to the purpose of the review. This can be overwhelming and may disengage readers who are already familiar with the basic pathology of UC.

Unclear Differentiation from Previous Work: While there are references to distinct aspects of UC, there is no mention of how this review differs from past reviews. A clearer explanation of its unique contribution—such as insights into emerging diagnostic tools, specific therapeutic targets, or a focus on precision medicine—would help clarify its relevance.

No Overview of Review Structure: The introduction would benefit from a brief roadmap, guiding the reader on the organization of the review and the main topics covered. This would help in setting clear expectations for readers.

Experimental design

The article places too much emphasis on basic information, with limited analysis or rationale, which impacts the clarity of its objectives and structure. To enhance the study design, the introduction should reduce redundant background content, focusing instead on critical knowledge gaps, recent advancements, and specific research questions that guide the review’s direction. This approach would improve coherence and ensure the review meets the standards of clarity and purpose expected for a specialist audience.

Validity of the findings

No comment

Additional comments

The article lacks a clear and logical flow of information. It primarily reiterates basic content that has already been covered in earlier reviews. There is a noticeable absence of new perspectives, scientific reasoning, or a comprehensive collection of relevant molecular findings that could contribute to advancements in diagnosis or treatment.

Reviewer 2 ·

Basic reporting

The introduction effectively sets the stage by defining UC as a serious complication of chronic kidney disease (CKD) and distinguishing it from other forms of cardiomyopathy. The emphasis on diastolic dysfunction and left ventricular hypertrophy (LVH) as primary features is well articulated, supported by relevant citations. The mention of UC's association with cardiac fibrosis and sudden cardiac death underscores its clinical significance, particularly in end-stage renal disease (ESRD) patients.

Experimental design

The study design is thorough, detailing a systematic literature search across multiple databases. The inclusion and exclusion criteria are clearly defined, ensuring the credibility of the selected studies. This section could benefit from a brief explanation of how the findings were synthesized, which would provide further clarity on the review process.

Validity of the findings

The article is well-researched and informative, addressing a critical area of concern in nephrology and cardiology. It successfully communicates the importance of understanding UC for improving patient outcomes in those with CKD.

Additional comments

Suggestions for Improvement
1. The article could benefit from a more detailed discussion on potential therapeutic strategies and interventions being explored for UC.
2. Including recent statistics or projections about the prevalence of UC could enhance the urgency of understanding and addressing this condition.
3. A clearer delineation of the implications for clinical practice, particularly in managing patients with CKD and UC, would add practical value for clinicians.

---

## Round 0.2 · accepted · Accept

The revised version of the manuscript is significantly improved. No further comments.